# Autotaxin/Lysophosphatidic Acid Axis: From Bone Biology to Bone Disorders

**DOI:** 10.3390/ijms23073427

**Published:** 2022-03-22

**Authors:** Candide Alioli, Léa Demesmay, Olivier Peyruchaud, Irma Machuca-Gayet

**Affiliations:** Univ Lyon, INSERM, Unit 1033, LYOS, F-69372 Lyon, France; candide.alioli@inserm.fr (C.A.); lea.demesmay@inserm.fr (L.D.); olivier.peyruchaud@inserm.fr (O.P.)

**Keywords:** autotaxin, lysophosphatidic acid, GPCR, bone, osteoblast, osteocyte, osteoclast

## Abstract

Lysophosphatidic acid (LPA) is a natural bioactive phospholipid with pleiotropic activities affecting multiple tissues, including bone. LPA exerts its biological functions by binding to G-protein coupled LPA receptors (LPA_1-6_) to stimulate cell migration, proliferation, and survival. It is largely produced by autotaxin (ATX), a secreted enzyme with lysophospholipase D activity that converts lysophosphatidylcholine (LPC) into active LPA. Beyond its enzymatic activity, ATX serves as a docking molecule facilitating the efficient delivery of LPA to its specific cell surface receptors. Thus, LPA effects are the result of local production by ATX in a given tissue or cell type. As a consequence, the ATX/LPA axis should be considered as an entity to better understand their roles in physiology and pathophysiology and to propose novel therapeutic strategies. Herein, we provide not only an extensive overview of the relevance of the ATX/LPA axis in bone cell commitment and differentiation, skeletal development, and bone disorders, but also discuss new working hypotheses emerging from the interplay of ATX/LPA with well-established signaling pathways regulating bone mass.

## 1. Introduction

Among the small lipids existing in eukaryotes, lysophosphatidic acid (LPA, 1-acyl-2-hemolytic-sn-glycerin-3-phosphate) is the smallest glycerophospholipid that has diverse biological functions such as control of cell growth, differentiation, cell motility, survival, and cytoskeleton change. In the 1990s, significant progress in the understanding of the mechanisms of action of LPA was made with the first identification of the type 1 LPA receptor. Since then, other receptors, mainly G-protein coupled receptors (GPCRs), have been identified, as well as their corresponding signaling pathways. All are rhodopsin-like type II receptors and have seven transmembrane domains, three extracellular loops (extracellular loop-ECL1, ECL2, ECL3), and three intracellular loops (intracellular loop-ICL1, ICL2, ICL3 [1]. At least four Gα subunits (Gα12/13, Gαq/11, Gαi/o and GαS) are used by the LPA receptors to signal, thereby activating different downstream pathways which under different environments and cell types give diverse biological outcomes [2]. Moreover, various LPA species have been isolated and identified in later years, that may differentially activate LPA receptor subtypes [3]. Numerous studies have shown that LPA and its receptors are crucial in multiple pathological mechanisms including neurological diseases, inflammation, tumors, metabolic diseases, and cardiovascular diseases [1,4,5]. More interestingly, the local presence of LPA in these systems has a significant biological impact. Autotaxin (ATX/ENPP2) is now considered to be responsible for the synthesis of the majority of extracellular LPA, though LPA can be produced from different lipid substrates. Autotaxin (ATX) has a plasma lysophospholipase D activity which produces LPA by hydrolyzing lysophosphatidylcholine (LPC) [4,5]. This enzyme, first identified as an autocrine motility-stimulating factor, was isolated from the supernatant of A2058 melanoma cells [6]. ATX genetic deletion and abrogation of LPA production result in embryonic lethality, due to aberrant vascular homeostasis and neural tube defects, indicating a major role for ATX/LPA in embryonic, vascular, and neuronal development [7,8,9]. It is highly expressed in the brain and reproductive organs and increased ATX levels have also been detected in human pregnancy, suggesting a role for ATX in reproduction [4].

Evidence that LPA is produced within bone tissue has been obtained in the context of bone metastasis where it acts as a paracrine factor stimulating cancer cell proliferation, cytokine secretion, and osteoclastic bone resorption [10,11]. Thus, in bone, adipocytes, osteoblasts, and osteoclasts may be potential sources of physiological LPA. Despite that, in vitro LPA is a serum-borne factor mandatory for RANKL-induced osteoclast formation [10,11], whereas, in vivo, the origin of LPA and the role of ATX in bone remain incompletely understood, especially during endochondral growth. The therapeutic use of LPA and LPA derivatives in bone regeneration has been proposed [12]. However, the levels of systemic LPA are highly complex, as most eukaryotic cells, including bone cells, express various forms of LPA receptors and present potentially differential ATX expression in young and old bone tissue [10,13,14,15,16]. As a consequence, activation of different cell types in bone may undermine the complex mode of action of LPA in bone pathophysiology and aging due to the pleiotropic activities of LPA through co-activation signals from multiple receptors. Understanding the role of each type of LPA receptor in bone cell function in situ is crucial for more effective therapeutic applications for both musculoskeletal disorders and the bone aging process.

In this review, we provide an overview of the current knowledge and perspectives of the emerging role of ATX and LPA signaling in bone development, bone lifespan, and aging. The therapeutic potential of an ATX pharmacological blockade in a pathological context will be discussed.

## 2. ATX/LPA Signaling in Healthy Bone

### 2.1. ATX/LPA and Growth Plate Formation

The proliferation, maturation, and organization of chondrocytes in growth plate cartilage is essential for the suitable longitudinal growth of bones. The ATX/LPA axis participates in the control of these cellular mechanisms. In 1998, Koolpe et al. were the first to link LPA and cartilage tissue when they showed that LPA was able to induce an increase in intracellular calcium in articular chondrocytes [17]. LPA was shown to stimulate rat articular chondrocyte proliferation; LPA_1_ and LPA_3_, but not LPA_2_, expression was reported in these cells. LPA-induced proliferation was almost completely blocked by PD98059 and Pertussis toxin (PTX a Gi protein inhibitor), suggesting an activation of the Gi/ERK-coupled signal [18]. There may be multiple origins of LPA in the growth plate microenvironment, but LPA derived from ATX activity may be an important source. Lysophosphatidylcholine (LPC) and lysophosphatidylethanolamine (LPE) are produced in rat growth plate chondrocytes in a maturation state-dependent manner [19]. For example, resting rat zone chondrocytes release LPA upon stimulation by 24,25-dihydroxyvitamin D3 [20,21,22]. Using rat knee joint models to mimic cartilage defects after acute injury, Wu et al. reported that ATX and LPA are highly produced at the site of injury by mesenchymal stromal cells and chondrocytes. In this model, LPA induces collagen type I expression in chondrocytes and bone marrow stromal cells through the Gi-PI3K pathway [23]. Pharmacological inhibition of the ATX/LPA axis by BrP-LPA reduces the collagen type I deposition at the site of joint injury in favor of the collagen II-enriched normal state [23]. Indeed, *Lpar1*-deleted mice are stunted with cranial dimorphism [24]. In addition to short stature, *Lpar1* KO mice present rib cage deformities associated with delayed vertebral calcification [25]. Nishioka et al. demonstrated, in homozygous *Lpar1* mutant zebrafish and the *Lpar1* KO mouse model, that loss of ATX-LPA_1_ signaling leads to dyschondroplasia with craniofacial malformation and impairment of cartilage formation [13]. Similar results have been obtained in the zebrafish ATX KO model in which ATX-LPA_1_ signaling acts by promoting S-phase entry and cell proliferation of chondrocytes through β1-integrin activation leading to fibronectin assembly and correct deposition of the extracellular matrix [13]. Taken together, these data suggest that the ATX/LPA/LPA_1_ axis controls—as an autocrine and paracrine factor—the formation of the growth plate and chondrocyte activity.

### 2.2. The ATX/LPA Axis: A Dual Effect on Bone Remodeling and Bone Formation

#### 2.2.1. Osteoclastogenesis and Modulation of Bone Resorption

Pharmacological targeting of osteoclasts, the bone resorbing cells, has always been a promising strategy to manage bone loss in many pathologies. In this context, basic research has revealed the implication of several cytokines in the control of bone resorption such as RANK-L, MCSF, IL6, and IL-1β. Interestingly, ATX/LPA was found to act on osteoclastogenesis and bone resorption. David et al. reported that, in vitro, osteoclast differentiation was enhanced by conditioned media from human breast cancer MDA-B02 cells with forced expression of ATX, or by recombinant autotaxin, and that this process was blocked by the autotaxin inhibitor vpc8a202. Furthermore, in vitro, the addition of LPA to active charcoal-treated serum was able to restore its capacity to support RANK-L/MCSF-induced osteoclastogenesis, suggesting the requirement of LPA during osteoclastic differentiation [11]. Moreover, in mature differentiated rat osteoclasts and RAW264.7-derived osteoclast cell lines, LPA promotes osteoclast survival via the nuclear translocation of NFATc1, effects that are blocked by VPC-32183 or by a specific peptide inhibitor of NFAT activation [26]. LPA has been also reported to promote phosphorylation of thyroid hormone receptor-interacting protein 6 (TRIP6) in RAW 264.7 cells [27] and activation of the non-receptor tyrosine kinase, c-Src, a key factor in the actin dynamics of osteoclast sealing zone structure, required for osteoclast resorptive function [28]. In murine bone marrow monocyte precursors, we found dynamic changes in the expression levels of LPA_1,2,4,5,6_ during osteoclastogenesis. All LPA receptors (except LPA_3_ and LPA_6_) are increasingly expressed in osteoclasts during the differentiation process. LPA_6_ expression is extremely high in bone marrow monocytes; even if it decreases during osteoclastogenesis, its level remains elevated in mature osteoclasts, whereas LPA_3_ is not detected [10] (Figure 1). We observed that in the presence of RANK-L and MCSF, monocytic precursors derived from the bone marrow of *Lpar1*^-/-^ mice exhibit a reduced capacity to form mature osteoclasts, unlike those of *Lpar2*^-/-^ and *Lpar3*^-/-^ mice. In addition, cultures of wild type monocytic precursors from bone marrow and mature osteoclasts, treated with the LPA_1/3_ antagonist Ki16425, showed impaired osteoclastogenesis. Moreover, the Ki16425 treatment of mature wild type osteoclasts seeded on bone slices alters actin sealing zone organization and significantly decreases the surface and number of resorption pits [10]. These latter results indicate that LPA/LPA_1_ signaling is mandatory for in vitro RANK-L and MCSF-induced osteoclastogenesis and osteoclast resorption activity. It is evident that this also occurs in vivo, especially since both genetic ablation and pharmacological inhibition of *Lpar1* remarkably alter the mineral matrix resorption of mature osteoclasts and prevent ovariectomy-induced bone loss [10,11]. However, in a previous study, Lapierre et al. found that LPA reduced rat osteoclast resorption activity, though this apparent discrepancy may be explained by a very different osteoclast isolation procedure. In this later study, mature osteoclasts were purified from rat or rabbit total long bones using bone cell differential adhesion properties [26]. Finally, a direct role of ATX in osteoclast function may occur during osteoclastogenesis and/or osteoclast activity. We recently demonstrated that mature osteoclasts produce functionally active ATX [16]. Taken together, these data show the importance of the ATX/LPA axis in osteoclast generation and functional activity that directly influences bone loss in several pathophysiological contexts, such as bone metastasis or inflammatory bone loss. These aspects will be discussed more in depth in the corresponding section of this review.

#### 2.2.2. Osteoblastic Lineage: From MSC to Full Mature Mineralizing Osteoblasts

Osteoblastic differentiation commences from multipotent bone marrow-derived mesenchymal stromal (stem) cell (BMSC) commitment in early osteoblast progenitor cells capable of differentiating into mature osteoblasts. Several molecules, including LPA, through their signaling pathway, control the commitment and maturation steps. *Lpar1* expression is high during the early steps of differentiation and remains stable in mature osteoblastic cells as reported for the human, rat, and mouse primary osteoblasts in UMR 106-01, G292, MG-63, and MC3T3-E1 osteoblastic cell lines [15,29,30,31]. LPA_1_ was found to be expressed in primary human BMSCs [31]. In contrast, LPA_2_ is low and LPA_3_ expression is scarcely reported in bone cells [29,31]. Moreover, *Lpar2* and *Lpar3*-invalidated mice did not display any bone impairment, suggesting that the activity of these receptors is dispensable in bone biology [32]. Interestingly, Liu et al. found LPA_4_ expression to be lower than LPA_1_ in human immortalized BMSCs (hBMSC-TERT), although LPA_4_ mRNA was increased in differentiated hBMSC-TERT [15]. In vivo, only LPA_1_ and LPA_4_ expression are detected in bone from 23-week-old mouse tibias, with a higher level for *Lpar4* than *Lpar1*. We found a relatively high mRNA level of LPA_1_ compared to LPA_2,3_ and LPA_4_ in 4-week-old mouse femurs [33]. This suggests a specific and differential regulation for *Lpar1* and *Lpar4* when comparing growth to aging. Figure 1A recapitulates LPA receptor patterns during osteoblast and osteoclast differentiation.

ATX and LPA are soluble molecules, both being found at high levels in the blood and serum. As a consequence, they are considered to be key soluble mediators. In vitro treatment of osteoblast cell lines with LPA induces very diverse responses relative to matrix synthesis, survival, proliferation, and mineralization.

Indeed, in mouse primary osteoblasts and SaOS-2 cells, LPA significantly impairs serum deprivation-induced apoptosis [34]. This anti-apoptotic cell response is blunted when GPCR-associated Gi protein pathway inhibitors are used in the assay, namely PTX, the Gi protein blocker, LY294002, and wortmannin for PI3K. In contrast, the LPA anti-apoptotic outcome is unaffected by Mek antagonists, indicating that osteoblast survival is indeed mediated by the LPA-PI3K-coupled pathway (Figure 2). Grey et al. also demonstrated that LPA promotes osteoblast proliferation through Gi and protein kinase C (PKC) activation [30]. There is growing evidence indicating that LPA and LPA receptors are involved in osteoblastogenesis. Mansell et al. found that LPA supports osteoblastogenesis and observed a deletory effect of Ki16425 on osteoblast differentiation [31].

Young *Lpar1*^-/-^ mice displayed an osteoporotic bone phenotype with a marked decrease in trabecular bone volume. Micro-CT analysis showed microarchitecture parameters were affected, with thinner, less numerous, and less connected trabeculae in the femurs and vertebrae, as well as a significant decrease in cortical bone thickness [25].

To gain further insight on the specific role of LPA_1_ in the osteoblastic lineage, we have recently generated conditional knockout mice for *Lpar1* using the osterix promoter Cre line in order to target early osteoblast precursor cells crossed with the *Lpar1^fl/fl^* line and generate *Lpar1*^ΔOb^ mice. The *Lpar1*^ΔOb^ bone phenotype does not fully recapitulate the strong bone impairment of *Lpar1*^-/-^ mice. Global *Lpar1* deletion affects the nervous system control of suckling, adipose cell development, and metabolic functions regulating glucose tolerance. All these abnormalities observed in *Lpar1*^-/-^ mice impact postnatal bone growth, hence, explaining the differences between the two genetically modified mice lines. Congruently, *Lpar1*^ΔOb^ conditional mutants reveal reduced bone mineralization and decreased cortical thickness, as well as increased bone porosity [33]. In our study, immortalized cl1-Ob-*Lpar1*^-/-^ osteoblasts revealed a remarkable cell proliferation defect associated with decreased YAP-P nuclear accumulation and reduced mineralization activity. *Lpar1* deficiency in osteoblasts leads to an alteration in osteogenic maturation reflected by increased expression of BSP, ALP transcription, and subsequent activity. Similarly, Liu et al. reported that, in osteogenic medium, LPA stimulates osteoblast differentiation of hBMSC-TERT which is also abrogated by a Ki16425 LPA_1/3_ inhibitor [15].

In contrast, downregulation of LPA_4_ by shRNA in hBMSC-TERT leads to increased mineralization, indicating that LPA_4_ is a negative modulator of the process. Supporting this concept, murine lines lacking *Lpar4* show a high bone mass phenotype as analyzed by micro-CT, demonstrating a marked increase in bone trabecular volume.

However, LPA_1_ and LPA_4_ appear to play opposing roles during osteogenesis, LPA favoring activation of osteoblast differentiation through LPA_1_ and inhibition through LPA_4_ down-stream signals [15]. In bone, BMSCs are common mesenchymal progenitors for osteoblasts and bone marrow adipocytes; commitment to one or the other lineage being the result of complex competitive molecular cascades. Basically, the osteoblast lineage is favored in young mice opposed to adipogenic lineage in aging mice. Xie et al. reported that *Lpar4* overexpression in stromal ST2 and preosteoblastic MC3T3-E1 cells inhibited osteogenic differentiation and, by contrast, enhanced adipogenic differentiation in ST2 and mesenchymal C3H10T1/2 cells [35]. These latter results highlight the importance of the LPA_1_/LPA_4_ interplay in BMSC to initiate commitment and differentiation in early osteoblastic or adipogenic lineages. Accordingly, *Lpar4*-deficient mice presented high trabecular bone volume and increased trabecular number and thickness [15]; moreover, Xie et al. reported that intratibial *Lpar4* silencing induces more ALP-positive osteoblasts on the trabecular bone surfaces, along with lower numbers and areas of adipocytes in the bone marrow.

In addition, LPA can downregulate PPARγ2 expression and thereby alter pre-adipose cell differentiation into adipocytes. Conversely, the anti-adipogenic effect of LPA was abolished in primary preadipocytes from mice invalidated for *Lpar1* which otherwise showed higher adipogenesis than the control mice fed with a high fat diet regimen [36]. Consistently, mice treated with Ki16425 are overweight and exhibit increased adipogenesis when compared to control mice fed with a high fat diet [37,38]. Taken together, in vitro data and analysis of global *Lpar1* or *Lpar4* knock-out mouse models underline the opposing roles for LPA_1_ and LPA_4_ in bone formation, resulting in a regulation of the directional differentiation of bone marrow mesenchymal stromal cells.

#### 2.2.3. Osteocytogenesis and Osteocyte Specification

Osteocytogenesis is the final step in osteogenic differentiation. Osteocytes represent over 90% to 95% of the total number of bone cells in the adult skeleton. They are ultra-differentiated osteoblasts completely embedded in the mineralized bone matrix. They have limited synthetic and resorptive capacities, but play a central role in maintaining bone homeostasis. Osteocytes are characterized by their cytoplasmic extensions or dendrites [39]. During osteoblast-osteocyte transition, the morphology of osteoblasts changes from a cuboidal shape to a more stellate shape which involves a vast reorganization of cytoskeletal proteins to promote the formation and elongation of dendrites [40,41]. E11 protein is a marker of osteocyte differentiation associated with dendritic extension [42]. E11 siRNA silencing blocks the dendritic process in MLO-Y4 osteocyte-like cell lines [43]. E11 activates the small GTPase RhoA, resulting in cytoskeleton rearrangement and increased cell motility [44], and FGF-2, in turn, by increasing E11 expression in osteoblastic cells promoting osteocytogenesis [45].

To date, very few studies have been performed on the specific role of LPA in osteocytogenesis. LPA was reported to induce dendritic outgrowth in MLO-Y4 osteocytes but not in the presence of Ki16425 or pertussis toxin, suggesting an LPA_1_/Gi pathway-dependent mechanism [46]. A proteomic study of LPA-induced dendrites in MLO-Y4 cells showed an enrichment in gene products associated with actin fiber dynamics, among which are E11 and Cx43, a major component of osteocyte gap junctions, also important for osteocyte survival [47,48]. We showed that osteocyte content and survival were markedly affected in long bones from *Lpar1*^ΔOb^ mice. Histological analyses of *Lpar1*^ΔOb^ bone tissue sections revealed an increase in the osteocyte apoptosis rate and larger areas of osteocytic lacunas. The osteocyte dendritic network imaged by confocal microscopy was also markedly reduced in the cortical bone of *Lpar1*^ΔOb^ mice compared to control animals [33]. In addition, a significant decrease in osteocyte specification markers, including E11, sclerostin, and Dkk1, was observed in *Lpar1*^ΔOb^ mice. Moreover, osteocyte dendritic connections were drastically reduced in primary cultures of *Lpar1*^ΔOb^ in response to FGF-2 (10 ng/mL), indicating that the action of FGF-2 during osteoblast-osteocyte transition requires the LPA_1_ receptor [33]. Indeed, FGF-2 treatment was found to induce MEK/ERK and PI3K/Akt signaling in osteoblastic cells. However, surprisingly, the MEK inhibitor UO126 was unable to block FGF-2’s ability to promote E11 protein expression, despite a significant reduction in ERK activation [45]. Similar results were observed upon inhibition of PI3K/Akt and p38 signaling. These latter results indicate that alternative pathways may cooperate with FGF-2 to enhance E11 expression and osteocyte formation, supporting the hypothesis that the ATX/LPA/LPA_1_ axis may be a potential candidate. Figure 2 recapitulates LPA receptor signaling and potential interacting pathways during osteoblast differentiation and osteocyte specification.

From left to right, the osteoblastic differentiation sequence from MSCs to osteocyte: in the upper part of the figure, the canonical transduction pathway downstream of LPA_1_ and LPA_4_ activated receptors are shown. At the bottom part, their consequent transcription activation: LPA signals directly through LPA_1_ and LPA_4_ receptors interplay with further cross-talk via active βcatenin (grey stone) and the FGFs signaling pathway (green lines). At early osteoblast stages, the LPA_1_ receptor is predominant and induces osteoblast commitment and differentiation via Gi signaling, activating early gene expression of ALP, Col1, and BSP (bottom part of the figure- represented by black arrows). Potential synergy with the ROCK1/Beta catenin pathway, leading to possible transcriptional co-operation to enhance transcription, is shown by the discontinuous green arrow. In full mature osteoblasts, LPA_4_ expression increases, and its downstream signaling prevails to block/control osteoblastic differentiation by ROCK1/Beta catenin inhibition (continuous red line). In osteocytes, LPA1 signaling is again predominant. LPA_1_ cooperates with FGF2 (continuous green arrow), and probably with FGF7, to promote E11, Dmp1, Dkk1 expression, and osteocyte specification (black arrows).

#### 2.2.4. The ATX/LPA Axis and the Wnt/Beta-Catenin Anabolic Pathway

ATX and LPA are important actors in bone development and bone mass control and their interaction with well-established bone anabolic pathways is worthy of discussion. The Wnt/β-catenin canonical pathway is the most predominant Wnt-signaling affecting bone homeostasis and skeletal development. It is essentially mediated by the binding of Wnt ligands (Wnt 3a, Wnt10b) to the receptor complex containing FZD (frizzled) and either LRP5 (low density lipoprotein receptor- related protein 5) or LPR6. In the absence of Wnt stimulation, cytoplasmic β catenin is phosphorylated by a “destruction complex” consisting of glycogen synthase kinase-3 (GSK-3), adenomatous polyposis coli (APC), and axin. It is then ubiquitinated and rapidly degraded by the proteasomal system to prevent its cytoplasmic accumulation. In contrast, Wnt stimulation suppresses GSK-3 activity and induces the cytoplasmic accumulation of β-catenin. The accumulated β-catenin translocates to the nucleus where it induces the expression of target genes, such as transcription factors of the Lymphoid Enhancer binding Factor (LEF)/T-cell Factor (TCF) family, by inactivation of associated co-repressors [49,50]. In addition to being a pivotal regulatory protein of the wnt pathway, β-catenin is linked to cadherin-based cell interactions. β-catenin binds strongly to N-cadherin which is highly expressed in the osteoblastic lineage. Targeted overexpression of N-Cadherin in osteoblasts in transgenic mice showed increased N-cadherin interaction with the cytoplasmic tail of the Wnt co-receptor LRP5. This binding leads to an increase in β-catenin ubiquitination, causing downregulation of the Wnt canonical pathway and consequent low bone mass [51]. Additional studies, mostly in cancer cells, have shown that LPA induces β-catenin activation. For example, in HCT116 and LS174T colon cancer cells, LPA enhances proliferation via transcription and nuclear translocation of β-catenin, phosphorylation of glycogen synthase kinase 3β, as well as transcriptional activation of Tcf/Lef. This signaling is mediated by *Lpar2* associated with protein Gq to activate PKC [52]. Later, in H19-7 cells, β-catenin activation by LPA was also shown, being induced via inhibitory phosphorylation of GSK3β by PKC, MAPK, and PKA, but not PI3K [53]. LPA disrupts N- cadherin junctions in ovarian cancer cells, releasing β-catenin [54,55] and, consequently, the unbound β-catenin may then feed into the Wnt pathway. Interestingly, LPA effects on β-catenin were reported to be linked to ATX in an autocrine loop. Reinforcing this concept, for example in fibrotic lung allografts-derived mesenchymal cells, LPA induces NFATc1 that, in turn, induces ATX expression as well as β-catenin. The use of inhibitors blunting LPA_1_ or ATX signaling was sufficient to reverse the fibrotic state and abrogate their ability to establish fibrotic lesions after adoptive transfer in vivo [56]. These data are appealing since β-catenin has been found to be activated in mesenchymal cells in the presence of LPA and is required for LPA-induced migration in vitro [57]. β-catenin activation in the lungs is also regulated via LPA_1_-dependent GSK-3β phosphorylation. In contrast, ATX seems to also be a direct target of Wnt/β-catenin signaling. In myoblast cells in which the β-catenin encoding gene Ctnnb1 was silenced by RNA interference, Wnt stimulation failed to induce ATX expression.

Taken together, these data strongly support the hypothesis that β-catenin is a pivotal regulatory protein linking the LPA/ATX axis to the Wnt pathway and highlight the importance of autocrine ATX secretion and downstream LPA_1_/GSK-3β signaling in lung fibrosis development and potentially other tissues. It can then be speculated whether ATX/LPA has a similar action during bone remodeling and whether it has any relevance in bone health.

As we have previously discussed, LPA_1_ and LPA_4_ receptors have often been suggested to have opposing actions on bone [15]. Interestingly, Xie et al. reported that LPA_4_ inhibits osteogenesis by blocking RhoA/ROCK1/β-catenin signaling [35]. Overexpression of ROCK1 stimulates osteogenic differentiation by activating the Wnt β-catenin pathway, decreasing adipocyte differentiation, and attenuating the inhibition of osteogenic differentiation caused by LPA_4_ [35]. Conversely, this leaves room for the hypothesis that the osteogenic effects of the LPA_1_ receptor potentially result from stimulation of the Wnt pathway.

## 3. ATX/LPA Signaling in Bone Pathophysiology

### 3.1. Bone Disorders, Age-Related Bone Loss and Bone Repair

ATX was recently identified as being involved in the pathogenesis of Fibrodysplasia ossificans progressiva (FOP). Hino et al. have recently identified mTOR signaling as a critical pathway for aberrant chondrogenesis and heterotropic ossification of mesenchymal stromal cells derived from FOP-iPSCs induced by Activin A [58]. They found ENPP2 (ATX) to be one of the most up-regulated, among the genes responding to activin induction in FOP-iPSCs, 24 h after chondrogenesis. They demonstrated that ATX/LPA is an upstream regulator that partially, but significantly, mediates the enhanced mTORC1 signaling in chondrogenic induction of FOP-iPSCs, making the Activin-A/FOP-ACVR1/ENPP2/mTOR axis consistent as a potential target in this disease [58]. More data were reported related to the involvement of ATX/LPA in bone loss. We have shown that LPA_1_ expression increases in osteoclasts derived from osteoporotic ovariectomized BALB/c mice and that treatment with Debio0719 (Ki16425 stereoisomer) and Ki16425 significantly prevents ovarierectomy-induced bone loss [10]. In contradiction with this data, Orosa et al. reported that C57BL/6OlaHsd ovariectomized mice treated with Ki16425 display similar results to vehicle-treated mice. However, in their experimental settings, the ovariectomized mice were too young (5 weeks old) and, in addition, the Ki16425 treatment regimen was discontinuous. These two elements easily explain the failure of Ki16425 to prevent bone loss [59]. Miyabe et al. showed that LPA_1_ antagonist, as well as LPA_1_ abrogation, also ameliorated murine collagen-induced arthritis CIA [60]. Both studies, ours and Miyabe’s, obtained with different experimental models, fully support functional control by LPA/LPA_1_ of osteoclast differentiation and bone resorption in vitro and in vivo, establishing LPA as a factor that may influence bone metabolism in menopausal women.

LPA appears to be a promising factor for bone regeneration in tissue engineering applications. Binder et al. showed that the survival of hBMSCs is enhanced in vivo by alginate hydrogels containing physically entrapped LPA [61]. Albumin-bound LPA cooperates with an active 1.25 OH vitamin to markedly increase the ALP activity of MG-63 osteoblast maturation on both Ti and hydroxyapatite (HA)-coated Ti surfaces [62]. This effect was also confirmed by Bosetti et al. in a three-dimensional (3D) collagen gel containing albumin-bound LPA and 1.25 OH vitamin D that presents a higher capacity for bone repair as an injectable scaffold, not only by promoting the maturation and migration of human osteoblasts but also by accelerating the apposition of bone fragments and new bone formation [63]. In addition, Yu et al., using a mouse model of osteotomy where the bone tissue was replaced with a degradable biomaterial, chitosan/beta-tricalcium containing either PBS, LPA, or an analog of LPA (monofluorinated and difluorinated LPA), allowed complete bone repair as indicated by micro CT and histological analyzes [12].

The development of LPA analogs may be promising as new methods for bone regeneration and wound healing, even though the real benefit in a human application may present challenges, depending on age, localization of the wound, or bio-disponibility of the LPA.

### 3.2. Inflammatory Bone Diseases

In inflammatory conditions, the balance between osteoblast-dependent bone formation and osteoclast-dependent bone resorption shifts toward bone resorption, leading to osteolytic bone lesions. These complications are observed in rheumatoid arthritis, spondyloarthritis, and periodontitis with systemic osteoporosis and increased fracture rates [64]. The ATX/LPA axis has been recognized to drive some mechanisms in these inflammatory bone contexts through its action on the inflammation process and associated bone loss.

The anti-inflammatory proteins sST2, ST2L, and heat-shock protein 25 (HSP25) were found by Affymetrix gene chip arrays to be the most highly upregulated among the 513 gene products significantly modulated by LPA in MC3T3-E1 osteoblast cell line after 6 h of treatment [65]. This supposes that LPA is able to modulate osteoblast function during inflammation [65]. However, the nature of this control could not completely be reduced to an anti-inflammatory action since LPA stimulates Interleukin 6 (IL-6) and IL-8, known not only to be two major proinflammatory cytokines but also as two potent inducers of osteoclast formation [54]. Bourgoin et al. reported that LPA- induced migration and secretion of IL-8/IL-6 in fibroblasts, such as synoviocytes (FLS), isolated from the synovial tissues of rheumatoid arthritis (RA) patients [66]. This effect was found to be p42/44 MAPK-, p38 MAPK-, and Rho kinase-dependent and suppressed by VPC32183, an LPA_(1/3)_ receptor antagonist [66].

Interestingly, LPA induced IL-8/IL-6 secretion in osteoblastic cells in a dose-dependent manner, this process being blocked by Ki16425, PTX, 2-APB, and U73122, treatment, suggesting that a potent LPA/LPA_1_ PLC transduction and IP3-mediated intracellular calcium activation occur in inflammatory conditions [67]. In addition, treatment of FLS with tumor necrosis factor-alpha (TNF-alpha) increases LPA_3_ mRNA expression in FLS and leads to the overproduction of cytokines in the presence of LPA [66]. This suggests a strong potential for LPA to impact proinflammatory crosstalk and bone loss during the inflammation process.

Beyond these numerous in vitro observations, the ATX/LPA/LPA_1_ axis raises growing interest in the pathophysiology of RA. The ATX level is elevated in synovial fluid from patients with RA and, in addition, LPA_(1/3)_ receptor antagonists and ATX inhibitors reduce the synovial fluid-induced cell motility [66]. The expression of LPA_1_ mRNA and LPA_1_ protein was found to be higher in FLS and synovium from RA patients than in FLS from osteoarthritis tissue [59,68]. Other mediators such as VEGF, CCL2, and MMP-3 were enhanced by LPA in RA FLS in an LPA_1/3_-dependent activation [68]. In addition, the silencing of LPAR1 abrogated TNF-induced proliferation and sensitized the RA FLS, but not the OA FLS, to TNF-induced apoptosis [59]. Indeed, mice with global deletion of *Lpar1* revealed a strong resistance to bone destruction in an arthritis model induced by type II collagen injection and showed decreased infiltration of macrophages and T helper cell differentiation into Th17, but not Th1 or Th2 [60]. Similar results were found with the pharmacological inhibition of LPA_1_, showing the influence of LPA/LPA_1_ on osteoclast differentiation and function in inflammatory osteolysis.

Similarly, pharmacological inhibition of the LPA_1_ receptor by Ki16425 in the K/BxN serum-transfer arthritis model led to a reduction in arthritis severity by decreasing cartilage damage, bone erosion, and expression of RANKL, ITAC/CXCL11, Pro-MMP9, and MMP3, in mouse arthritic joints [69].

Surprisingly, we were able to show that the ATX conditional mutant in osteoclast (ΔATX^Ctsk^) mice displays no effect on bone mass at a steady state [16]. However, ATX is strongly induced in inflammatory osteoclasts at erosion sites in bone joints from hTNF^+/−^ transgenic mice. In addition, when ΔATX^Ctsk^ transgenic mice undergo several inflammatory challenges, either by LPS treatment, K/BxN-serum transfer, or ATX antagonist pharmacological treatment in the hTNF^+/−^ mouse, then the systemic and local inflammatory bone loss was almost fully prevented. We, therefore, conclude that ATX is a novel functionally active autocrine factor produced by osteoclasts that specifically controls inflammation-induced bone erosion by osteoclasts and systemic bone loss [16]. Taken together, these data suggest that blockade of ATX activity either genetically or pharmacologically prevents systemic bone loss and bone erosion, offering a novel therapeutic approach for RA patients.

### 3.3. Bone Metastasis

The ATX/LPA axis has been reported to regulate cell proliferation, migration, and survival in many cancer cells and as a consequence is recognized as a major regulator of tumorigenesis, angiogenesis, and the metastatic process [70,71]. Increased ATX and LPA levels are found in different types of cancer cells [72,73]. In cancer patients, bone metastasis is one of the most common complications that occur, and high ATX expression in primary tumors is frequently associated with a poor prognosis [74,75,76]. Interestingly, endogenous expression of ATX in 4T1 mouse carcinoma cells or high ATX expression in human MDA-BO2-ATX-transfected breast cancer cells was reported to provide a higher propensity for these cells to generate bone metastasis [11]. A possible mechanism of ATX/LPA-promoted bone metastasis involves the promotion by LPA of cancer cell motility and proliferation, as well as the secretion of growth factors such as VEGF and cytokines such as interleukin IL-6, IL-8, GMC-SF, and MCP-1 that are known to affect bone remodeling [77,78]. Overexpression of LPA_1_ in MDA-BO2/LPA_1_-transfected cells dramatically increases the extent of osteolytic lesion areas, whereas pharmacological inhibition of this receptor, using Ki16425 and Debio 0719, reduced cytokine production and the progression of osteolytic bone metastasis. Intriguingly, endogenous expression of ATX was revealed to be dispensable for cancer cells to metastasize to bone, as shown by the above-mentioned MDA-BO2 cells that do not express ATX [79]. Our lab reported that challenging mature osteoclasts with inflammatory molecules such as TNF and LPS upregulates the secretion of ATX through an NF-κB pathway [16]. Furthermore, during our study on the role of non-tumoral ATX in the formation of bone metastases, we found that blood platelets uptake circulating ATX that is naturally present in the bloodstream and store ATX in their granular compartments which is eventually released under tumor cell-induced platelet aggregation [80]. Non-tumoral ATX released by platelets is functionally active as it catalyzes the production of LPA which ultimately acts on cancer cells to promote survival, invasion, and bone metastasis. Treatment with an ATX inhibitor (BMP22) of mice harboring pre-established bone metastases from the ATX-null MDA-BO2 breast cancer cells significantly reduced the progression of osteolytic lesions [80,81]. Moreover, we showed in a recent study that the production and activity of LPA following tumor cell-induced platelet aggregation constitutes a second vicious cycle between tumor cells and blood platelets taking place at the bone metastasis site [81]. Non-tumoral ATX released by activated platelets is likely to contribute to this bone resorption-independent cycle. Taken together, ATX present in the tumor environment appears to be a major promoter of tumor growth and metastasis and bone resorption, representing a new aspect of the multiple interconnected vicious cycles established at the level of bone metastasis [82].

### 3.4. Bone Pain

Bone pain behavior has been mostly investigated in the context of bone metastasis and bone inflammatory disorders. The former depends first on the consequences of exacerbated bone resorption as shown primarily by Yoneda et al.; proton release during osteoclast-mediated resorption creates acidic microenvironments that stimulate sensory nociceptive neurons innervating bone [83]. That is why anti-resorptive agents, such as bisphosphonates and Denosumab, efficiently reduce bone pain in bone metastasis patients. As mentioned previously, LPA promotes stimulation of osteoclast activity and therefore could indirectly generate metastasis-induced bone pain. However, using an osteosarcoma-induced bone pain model in the rat, Zhao et al. [84] showed that sural C-fibers become sensitive to LPA stimulation, an effect that is blocked by treatment with VPC32183, an antagonist of the LPA_1_ receptor, indicating a direct action of locally produced LPA. In addition, two weeks after cancer cells were injected into rat tibia, analysis of dorsal root ganglion neurons revealed increased LPA_1_ receptor expression [85]. In this case, bone pain was conveyed from the dorsal root ganglion neurons through vanilloid 1 (TRPV1) receptors linked to protein kinase Cε and potentiated by LPA through LPA_1_ signaling.

Secondly, rheumatoid arthritis (RA)-associated pain is not a simple outcome of inflammation because even though DMARDs (disease-modifying antirheumatics drugs) are able to control and resume joint inflammation, treated patients continue to suffer from persistent pain [86]. In a rat model of osteoarthritis, McDougall et al. showed that intra-articular injection of LPA caused knee joint neuropathy, joint damage, and pain. In this case, the pharmacological blockade of LPA receptors inhibited joint nerve damage and hindlimb incapacity. These results underline the LPA pathway as an emerging molecular neuropathic component of bone pain [87].

Moreover, Collagen Antibody Induced Arthritis (CAIA) mice immunized with collagen type II antibodies, develop an autoimmune response and joint pathology similar to that observed in human RA [88]. These mice display a first phase of neuropathic pain during joint inflammation but also in a later phase after the resolution of inflammation. Very recently, Su et al. demonstrated that the neuropathic-like phase of CAIA is reversed by a neutralizing antibody generated against LPA and by an LPA_1/3_ receptor inhibitor-without affecting joint inflammation [89]. Analysis of dorsal root ganglia revealed profound neurochemical changes and showed that ATX is upregulated in the DRG neurons during both pain phases. Within the dorsal root ganglia, mouse and human satellite glia cells were identified as cells expressing activated LPA_1_ receptor, leading to increased production of pronociceptive factors such as cytokines or NGF. In turn, pronociceptive factors promote an increase in ATX levels as well as nociceptor excitability with prolonged sensitization in sensory neurons which leads to chronic pain. This work definitively demonstrates the role of ATX/LPA/LPA_1_ signaling in RA-associated pain [89].

## 4. Conclusions

Within the two last decades, the ATX/LPA axis has emerged as a key component of bone pathophysiology. To date, *Lpar1* and *Lpar4* null models—despite associated perinatal mortality—have helped to understand ATX/LPA physiological functions in cartilage and bone cell proliferation, differentiation, and activity.

Although osteoblast conditional mutants of *Lpar1* certainly display a more discrete bone phenotype, they have contributed to delineating the specific role of LPA_1_ in osteoblast late differentiation and osteocyte specification [33]. More studies involving the conditional mutants of *Lpar1* and *Lpar4* in the full panel of bone cell type are needed to better dissect the biological function of LPA signaling in bone remodeling. In contrast, ATX null and ATX overexpressing genetically modified animal models have demonstrated how critical ATX is for embryological development since they are lethal or show severe vascular and neural crest defects and, more recently, early osteo-chondrogenesis abnormalities.

Strikingly, ATX heterozygous mice or osteoclast ATX conditional deletion mutants do not have an effect on either bone mass or bone remodeling, nor other tissues for the former model. In contrast, in a pathological preclinical model for autoimmune or inflammatory arthritis, we have shown that ATX is an autocrine factor produced by osteoclasts that specifically controls inflammation-induced bone erosion by osteoclasts and systemic bone loss [16]. Additionally, previous work using several pre-clinical models affecting glucose homeostasis, obesity-related insulin resistance, and idiopathic pulmonary fibrosis has shown the major importance of ATX/LPA local production by the specific cell type involved in the pathology [90]. Indeed, taken together these results are in line with the idea that, although all the biological functions of ATX are mediated by LPA, LPA is very sensitive to inactivation by lipid phosphate phosphatases and requires cell-surface-bound ATX for efficient LPA receptor activation. In this case, ATX is working as a docking molecule for LPA in addition to its lysophospholipase D activity to produce LPA from LPC. This concept is supported by several studies including ours, showing that ATX binds to the cell surface through its interaction with adhesion molecules to efficiently deliver LPA to its cognate LPA receptor [91]. This new mechanism offers better specific therapeutic alternatives to classical ATX/LPA inhibitors targeting either ATX Lysophospholipase D activity or blocking largely LPA receptors with limited specificity. Some of these inhibitors are indeed very efficient in preclinical models to sustain ATX inhibition or an LPA receptor blockade but have not passed phase 3 in clinical trials due to the pleiotropic effects of ATX/LPA signaling and unintended off-target consequences [92].

## Figures and Tables

**Figure 1 ijms-23-03427-f001:**
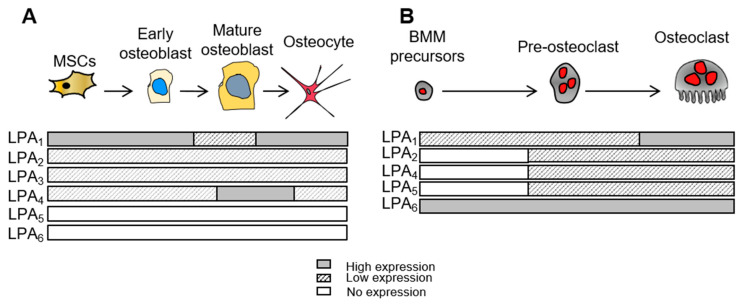
LPA receptors patterns in bone cells: (**A**) Osteoblast differentiation and osteocyte specification and (**B**) Monocyte-derived osteoclast differentiation.

**Figure 2 ijms-23-03427-f002:**
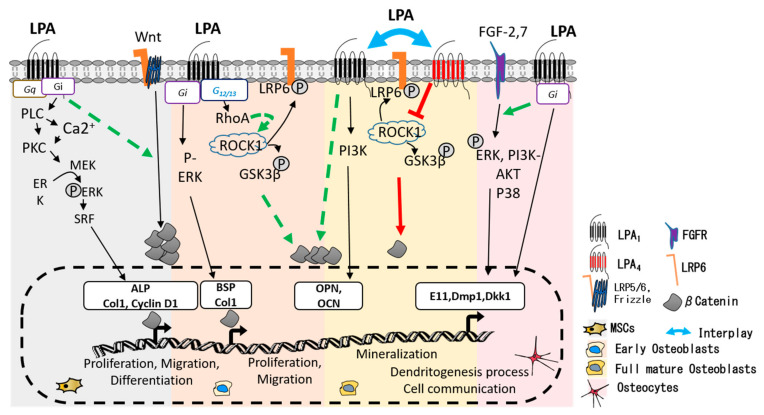
ATX/LPA signaling insight in osteoblastic lineage. From left to right, osteoblastic differentiation sequence from MSCs to osteocyte: in the upper part of the figure the canonical transduction pathway downstream of LPA_1_ and LPA_4_ activated receptors are shown. At the bottom part their consequent transcription activation: LPA signals directly through LPA_1_ and LPA_4_ receptors interplay and via further cross-talk with active βcatenin (grey stone) and FGFs signaling pathway (green lines). At early osteoblast stages, LPA_1_ receptor is predominant and induces osteoblast commitment and differentiation via Gi signaling activating early gene expression ALP, Col1, BSP (bottom part of the figure- represented by black arrows). Potential synergy with ROCK1/Beta catenin pathway leading to possible transcriptional co-operation to enhance transcription is shown by discontinuous green arrow. In full mature osteoblasts, LPA_4_ expression increases and its downstream signaling prevails to block/control osteoblastic differentiation by ROCK1/Beta catenin inhibition (continuous red line). In osteocytes, LPA1 signaling is again predominant. LPA_1_ co-operate with FGF2 (continuous green arrow) and probably with FGF7 to promote E11, Dmp1, Dkk1 expression and osteocyte specification (black arrows).

## Data Availability

Review not applicable.

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
