# Peer review of "Autotaxin/Lysophosphatidic Acid Axis: From Bone Biology to Bone Disorders"

_ijms, 2022, doi:10.3390/ijms23073427_

Round 1

Reviewer 1 Report

The manuscript “Autotaxin/Lysophosphatidic Acid Axis: from Bone Biology to 2 Bone Disorders” present an interesting review but need to be improve before publication

The manuscript present interesting data and the figures well illustrate the reported mechanisms, however before publication the manuscript need to be improve.

The manuscript need important English editing. It must be proofread by a native English

Most of the section are too long, too many detail about the experiments are included. The authors present a review they should better synthetize the informations to better highlight the data.

Author Response

We thank the reviewer 1 for appreciating our review and for the helpful comments that we have taken in account.

First the manuscript has been proofread by an external service  provided by  a native English Dr Robin Buckland PhD .

Second, long sections as 2.2.2Osteoblastic lineage: from MSC to full mature mineralizing osteoblasts “ was  reduced (or reorganized.  Each section is ending with a short sentence highlighting the overall message.  The modifications are as follows:

  • Line 159 to 162 suppressed and line 162-163 rephrased
  • To facilitate reading a paragraph starting from line 206 to 219, reporting several in vitro studies on LPA signaling on BMSCs was suppressed.  
  • Then in the same section information was reorganized focusing mainly on genetically modified mice models, involving Lpar1 and Lpar4 rather than in vitro studies. Lane 275 to 290.
  • Lines 293 to 401 were removed, however the information content was grouped line-275 with the part reporting LPA involvement in MSCs fate: osteogenic or adipogenic and opposed roles for LPA1 and LPA4 in this process.
  • Missing Figure 2 legend was added
  • In section3.1 ATX /LPA signaling in Bone pathophysiology; few lines reporting LPA / FGF signaling in osteocytes were removed as they were redundant with section 2.2.3.
  • Section bone repair was reduced from line 666 to 670.
  • The graphical summary was removed according to editing editor request.

Reviewer 2 Report

This is very high quality article and I enjoyed reading it. The authors have collected together a range of relevant material, it is well described, and contains many excellent figures. It will make a useful contribution to the field and is an excellent resource for those new to the field. There are a few typographical errors but overall the article is well written.

Author Response

We are extremely pleased that reviewer 2 enjoy reading our review. We warmly thank the reviewer for his comments. We really felt that the review  will be useful as up to now bone pathophysiology under the ATX/LPA axis view has never  been edited . The manuscript has been proofread by a Native English. Nevertheless, some revisions have been made:

  • We have greatly modified section section as 2.2.2 Osteoblastic lineage: from MSC to full mature mineralizing osteoblasts “ to facilitate reading.
  • We have also added a missing legend for Fig2. and removed the graphical summary according to editing editor request.
